# Asymmetric limits on timely interventions from noisy epidemic data
Kris V. Parag [1,2] ✉, Ben Lambert[3,4], Christl A. Donnelly [3,4] & Sandor Beregi[1]

Deciding on when to initiate or relax an intervention in response to an emerging infectious disease is both difficult and important. Uncertainties from noise in epidemiological surveillance data must be hedged against the potentially unknown and variable costs of false alarms and delayed actions. Here, we clarify and quantify how case under-reporting and latencies in case ascertainment, which are predominant surveillance noise sources, can restrict the timeliness of decision-making. Decisions are modelled as binary choices between responding or not that are informed by reported case curves or transmissibility estimates from those curves. Optimal responses are triggered by thresholds on case numbers or estimated confidence levels, with thresholds set by the costs of the various choices. We show that, for growing epidemics, both noise sources induce additive delays on hitting any case-based thresholds and multiplicative reductions in our confidence in estimated reproduction numbers or growth rates. However, for declining epidemics, these noise sources have counteracting effects on case data and limited cumulative impact on transmissibility estimates. We find that this asymmetry persists even if more sophisticated feedback control algorithms that consider the longer-term effects of interventions are employed. Standard surveillance data, therefore, provide substantially weaker support for deciding when to initiate a control action or intervention than for determining when to relax it. This information bottleneck during epidemic growth may justify proactive intervention choices.

Rapidly and reliably detecting the key dynamics of infectious diseases is a recurrent challenge in epidemiology with critical consequences and trade-offs[1–3]. Early warnings of upcoming epidemic growth or decline can provide valuable signals for mobilising or relaxing interventions[4–6] and even enhance the effectiveness of decisions to act[7]. However, epidemic data are often scarce or uncertain in early periods[8,9] and the economic and other costs of premature action (i.e., false alarms) may be detrimental. Choosing to wait for data to accumulate and improve the evidence base for decision-making can also be costly, as the resulting delays to intervene (i.e., missed actions) may mean lost opportunities to minimise epidemic burden or relax interventions when infections are, respectively, rising or waning[10–12].

Although the decision to act or not at any time commonly involves many external factors (e.g., political and sociocultural precedents)[13,14], at the core of these choices is the complex but fundamental trade-off between uncertainty in epidemic dynamics and uncertainty in the costs of deciding to intervene (and hence modify those dynamics). While integrated epidemiological and economic modelling frameworks[15–18] have been proposed to balance uncertainties and optimise decision-making, these approaches require knowledge or at least assumptions about these uncertainties, which can be difficult to quantify accurately. Even if this uncertainty is well-described, recommended actions may be sensitive to parameter, model structure, and implementation choices[7,19]. The complexity of integrated models may further constrain the ability to validate the robustness of outputs, precluding generalised insights[9,20,21].

Given these challenges, a complementary set of studies have instead aimed to use simplified models, which are easier to parametrise and validate, to uncover the fundamental limits that uncertainties impose on public health decision-making[22]. While these studies have yielded valuable insights into the robustness and timeliness of intervention choices and into how we can optimally detect shifts in epidemic dynamics[6,7,23–25], few works have directly coupled these to costs of missed actions and false alarms (though they do examine proxies e.g., the times over which interventions are sustained) or characterised how noise in surveillance data impacts detection and hence decision timepoints, which then feed back onto those costs. Here, we aim to resolve these gaps by adapting optimal Bayesian detection theory[26,27], with the goal of extracting general guidelines about how noise constrains cost-optimal action points.

[1]MRC Centre for Global Infectious Disease Analysis, Imperial College London, London, UK. [2]NIHR HPRU in Behavioural Science and Evaluation, University of Bristol, Bristol, UK. [3]Department of Statistics, University of Oxford, Oxford, UK. [4]Pandemic Sciences Institute, University of Oxford, Oxford, UK. ✉e-mail: k.parag@imperial.ac.uk

We show that the costs of missed actions and false alarms, even when not known accurately, imply a threshold criterion that we must compare with our evidence to act. We then describe how the information supporting this decision to act accumulates above this threshold with time. The costs of erroneous actions are therefore directly balanced by the aggregate of information that supports action, i.e., larger costs require more evidence and longer wait-times to act are justifiable[26]. For growing and waning epidemics, this evidence, respectively, informs the initiating and relaxing of interventions. Our central contribution is to reformulate this decision-cost framework to understand how under-reporting of infections and delays in reporting modify the optimal thresholds for supporting action. Interestingly, we find these ubiquitous sources of noise intrinsically but asymmetrically limit our capacity to respond to epidemics.

When an epidemic is growing, delays and under-reporting introduce additive lags on achieving any threshold based on the incidence of infections or related curves such as those of cases and deaths, and multiplicative reductions in information available for estimating transmissibility, i.e., uncertainties around estimates of reproduction numbers and growth rates are amplified by both noise sources. However, when an epidemic is waning, the noise sources counteract, with reporting delays offsetting the impact of under-reporting when hitting thresholds, and both have minimal effect on transmissibility estimate uncertainties. This asymmetry results from the directionality of delays and under-reporting, is practically magnified because data are often scarcer during early epidemic growth stages[23], and persists even if sophisticated decision-algorithms[28] that optimise intervention timings and their longer-term impact are applied. We argue that markedly more restrictive bottlenecks to optimal decision-making during emergent epidemics fundamentally support proactive outbreak intervention choices[9].

## Results

### Detection thresholds and the costs of intervening

We start by adapting Bayesian decision theory to better understand what a decision involves and how the impact of costs (even if not accurately known) can be included[29]. Given some information at time $t$, denoted $X_t$, on the dynamics of an infectious disease, we must decide if it is optimal to act now or wait for future information. We use $H_1$ to represent the hypothesis that we should act now and $H_0$ as the null hypothesis that we should not act. For a growing epidemic, this action may be to initiate non-pharmaceutical interventions (NPIs), while for a declining one, this may involve relaxing any existing NPIs. To determine what optimal means, we must attach costs to actions. We explore the mathematical details of this approach in the Methods, but here outline the essential components and results.

A missed action represents when we act too slowly and is a false negative (FN) with cost $c_{FN}$. A false alarm describes a premature action and is classed a false positive (FP) with cost $c_{FP}$. A true positive (TP) and true negative (TN), respectively indicate when action (or inaction) is appropriately taken, with associated costs of $c_{TP}$ and $c_{TN}$. Applying the principles of decision theory we can express the optimal binary decision at time $t$, $i_t^*$, as in Eq. (1) below.

$$i_t^* = \mathbf{1}\left(\mathbf{P}\left(H_1|X_t\right) \geq \eta = \frac{c_{FP} - c_{TN}}{(c_{FP} + c_{FN}) - (c_{TP} + c_{TN})}\right), \; t^* = \underset{t \geq 0}{\operatorname{argmin}}(i_t^* = 1).$$
(1)

Here $\mathbf{1}(.)$ is an indicator function. When its condition is satisfied, $i_t^* = 1$ and we should act. The optimal time of action is $t^*$, and $\eta$ is a threshold that depends only on relative costs.

This expression reveals that our decision to initiate (or relax) an intervention in the face of a potentially growing (or waning) epidemic rests on comparing the posterior evidence for the need to act, $\mathbf{P}\left(H_1|X_t\right)$, against the proportion of the total excess cost of making incorrect choices that results from false alarms. The time at which we are statistically justified in

acting is $t^*$ i.e., the first time when $\mathbf{P}\left(H_1|X_t\right)$ exceeds $\eta$. Larger $c_{FP} - c_{TN}$ is only counterbalanced by stronger evidence for the need to act and increases $t^*$. Equation (1) is noteworthy as whatever the costs of action or inaction, ultimately optimal decision-making involves testing a threshold $\eta$ against our evidence, which (assuming no change in the epidemic state) accumulates with time. When false alarms (or missed actions) are prohibitively more expensive then $\eta$ rises to 1 (or falls to 0) and the optimal time to act $t^*$ becomes infinite (or 0).

Since we cannot control or sometimes even know (as quantifying the costs of interventions is non-trivial) the value of $\eta$, we focus on understanding the factors that regulate the posterior evidence. If $H_1$ is true, then we expect that any threshold will eventually be crossed. However, for fixed costs, the lag to crossing is regulated by at least two key factors. The first is the choice of information source $X_t$. In the next section, we consider two widely used sources – the reported count of symptomatic cases over time and the inferred transmissibility parameters underpinning those cases. Second, the evidence from these sources for initiating (or relaxing) an intervention will deteriorate with the level of noise in the epidemic surveillance data. We examine and quantify, for the first time to our knowledge, how common types of surveillance noise reduce the information in these sources for guiding intervention decisions.

We further interpret this Bayesian decision problem using information theory. Effectively, we want to communicate a 1 or 0 to a policymaker to indicate the evidence, respectively, for action or inaction. When $H_1$ is true, then $\mathbf{P}\left(H_1|X_t\right) = p_t$ accrues with time and is the probability that we should act given the epidemic state (which we estimate from $X_t$). In Eq. (6) of the "Methods" we find Eq. (1) implies that $i_t^* = \mathbf{1}\left(-\frac{d\mathcal{H}_{p_t}}{dp_t} \geq \log_2 \frac{\eta}{1-\eta}\right)$ with $-\frac{d\mathcal{H}_{p_t}}{dp_t}$ as the loss in entropy as $p_t$ increases. Our decision-making process compares this loss or equally the increase in certainty that $H_1$ is true, against a cost-based threshold. The optimal time $t^*$ that a policymaker should wait before acting is controlled by how $p_t$ rises with time[30]. For any cost structure, this time depends on the surveillance noise, which reduces $p_t$ (when $H_1$ is true) and hence the optimality and speed of possible intervention decisions. We explore these effects next.

### Responding to epidemic growth and decline using reported incidence

The most timely and visible indicator of the likely state of an epidemic is the reported incidence of symptomatic cases, $C_t$ at time $t$. This serves as a proxy for the count of new infections, $I_t$, which is rarely observable. Case incidence also measures potential epidemic burden because the incidence of hospitalisations or deaths may be described as delayed and scaled versions of $C_t$[31,32]. A common approach to epidemic detection involves sequentially comparing the reported incidence against some baseline threshold[6,25,33,34]. We denote such a baseline by $a$ and define our decision problem with the posterior evidence of $\mathbf{P}\left(H_1|X_t\right) \equiv \mathbf{P}\left(I_t \geq a|C_1^t\right)$ for a growing epidemic and $\mathbf{P}\left(H_1|X_t\right) \equiv \mathbf{P}\left(I_t \leq a|C_1^t\right)$ for a waning or controlled epidemic. We use $Y_i^j$ for some variable $Y$ to represent the time series or vector $\{Y_i, Y_{i+1}, \ldots, Y_{j-1}, Y_j\}$. Our evidence is based on the likely number of infections given observed cases.

We start by deriving analytic insight from a related deterministic model of epidemic spread. As cases suffer from under-reporting and delays in reporting (relative to the underlying infections) we define a fixed reporting fraction $\rho$ (the proportion of infections reported as cases) and delay $\tau$ (the lag before an infection is reported as a case). We can then describe the true infection incidence $I_t$ and the reported case incidence $C_t$ as in Eq. (2) with exponential dynamics. Here $r$ is the effective growth rate, which corresponds to an effective reproduction number $R$ and $I_{ic}$ is the initial condition on the count of infections at time $t_{ic}$. If we are early in the epidemic, where susceptible depletion is negligible and with $t_{ic} = 0$, then $R = R_0$ and $r = r_0$ i.e., they become the basic reproduction number and intrinsic

**Fig. 1 | Extrinsic detection case threshold performance under noisy surveillance.** We plot the lag in crossing alert threshold $a$ for 1000 epidemics simulated with COVID-19 parameters under a renewal process (see Methods). **a** Upward crossing lag times $\Delta_{\text{grow}} = t_{C \geq a} - t_{I \geq a}$ during the growth stage of the simulated epidemics. **b** Downward crossing lags $\Delta_{\text{wane}} = t_{I \leq a} - t_{C \leq a}$ for waning epidemics. We plot lags for cases corrupted by only under-reporting (red), delays (green) and then for both noise sources combined (blue).

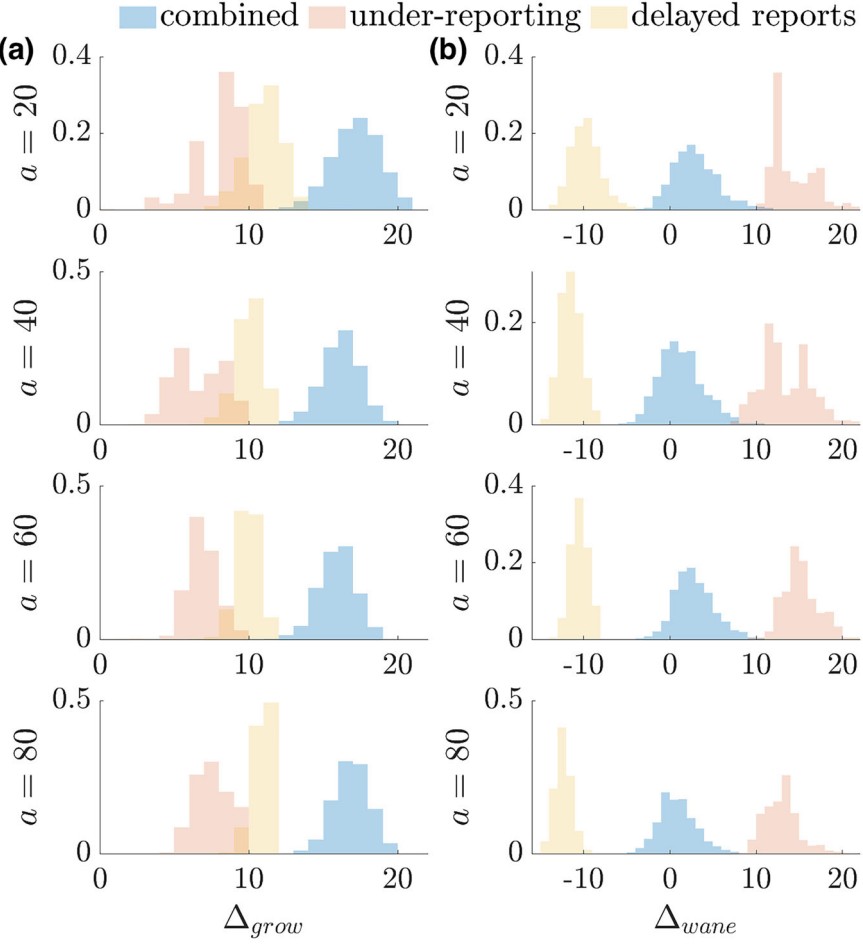

growth rate, respectively.

$$I_t = I_{\text{ic}} e^{r(t - t_{\text{ic}})}, \quad C_t = \rho e^{-r\tau} I_t, \quad r = \gamma(R - 1). \quad (2)$$

Here $\rho$ encodes effects such as under-ascertainment and asymptomatic spread, while $\tau$ describes latencies due to surveillance limitations and lags in diagnosing symptomatic patients. The last relation in Eq. (2) specialises this growth process to the dynamics of a classical susceptible-infected-recovered (SIR) or susceptible-exposed-infected-recovered (SEIR) model when the fraction of susceptible individuals is constant, with $\gamma$ as the duration of infectiousness. Equation (2) also approximates epidemic dynamics more generally as if the growth rate is time-varying (e.g., during declining epidemic stages), we can interpret $r$ as the mean $\frac{1}{t - t_{\text{ic}}} \int_{t_{\text{ic}}}^{t} r(s) ds$ and apply similar expressions to obtain averaged (approximate) reproduction numbers[35,36].

We assume that neither the reporting fraction nor the delay is known. This is common during the early stages of an emerging outbreak but may remain problematic into the later epidemic stages due to interventions or new pathogen variants changing the value of these and related epidemiological variables[37]. Consequently, we may often have to make decisions by simply choosing some alert threshold $a$ and finding the first time that this value is exceeded by the epidemic curve. Such exceedance approaches have been practically used[6] but, as far as we can tell, the heterogeneous impacts of major surveillance noise sources on these decision-making problems has not been quantified analytically. We consider both growing and waning epidemics with $\Delta_{\text{grow}} = t_{C \geq a} - t_{I \geq a}$ as the lag in exceeding our alert threshold for a growing epidemic and $\Delta_{\text{wane}} = t_{I \leq a} - t_{C \leq a}$ as the related time lag for a waning epidemic.

We calculate these crossing times by equating expressions from Eq. (2) with the alert value i.e., we solve for $a = I_{\text{ic}} e^{r(t_{I \geq a} - t_{\text{ic}})} = \rho I_{\text{ic}} e^{r(t_{C \geq a} - t_{\text{ic}} - \tau)})$. This yields Eq. (3) with $r > 0$ ($r < 0$) for growing (waning) epidemics with $\tau_\rho$ defined as an effective lag resulting from $\rho$.

$$\Delta_{\text{grow}} = \frac{1}{|r|} \log \frac{1}{\rho} + \tau, \quad \Delta_{\text{wane}} = \frac{1}{|r|} \log \frac{1}{\rho} - \tau, \quad \tau_\rho \overset{\text{def}}{=} \frac{1}{|r|} \log \frac{1}{\rho}. \quad (3)$$

While Eq. (3) is trivial to derive it embodies some key insights. First, we can treat the influence of incomplete reporting as an effective shift of $\tau_\rho$ time units. Second, $\tau_\rho$ adds to the reporting delay $\tau$ for growing epidemics but subtracts from it when aiming to detect waning epidemics. Accordingly, early detection of emerging epidemics is substantially more difficult than the early detection of waning epidemics. For example, if $\tau_\rho \approx \tau$, then $\Delta_{\text{grow}} \approx 2\tau$ but $\Delta_{\text{wane}} \approx 0$. This occurs when $\rho \approx e^{-|r|\tau}$, which yields $\rho = \frac{1}{2}$ if $\tau = \frac{\log 2}{|r|}$ equals the doubling (halving) time.

The asymmetry in Eq. (3) is striking and implies that a substantial bottleneck exists when responding to growing epidemics versus waning ones. Importantly, we find that this insight holds for more realistic models. We simulate epidemic case incidence according to the noisy renewal model from Eq. (8) of the Methods, which includes stochasticity both in the incidence generation and the reporting and delay distributions. Using parameters for COVID-19 we showcase the expected asymmetry for hitting alert thresholds in Fig. 1. Our results do not depend on $a$ being constant so we can also allow it to vary with time. In this setting, it can theoretically encode stochastic, time-varying and seasonal baselines for exceedance alerts.

The histograms of Fig. 1 summarise the lags in times at which cases $C_t$ exceed $a$ relative to infections $I_t$ i.e., $\Delta_{\text{grow}}$ and $\Delta_{\text{wane}}$. These inform on our evidence criteria $\mathbf{P}(I_t \geq a | C_1^t) \geq \eta$ for growing epidemics and

**Fig. 2 | Intrinsic reproduction number or growth rate detection performance under noisy surveillance. a** Presents ratios of the Fisher information for cases $\mathbf{I}(R|C_1^t)$ to that from the true infections $\mathbf{I}(R|I_1^t)$ for the growing phase (red, the true reproduction number $R > 1$ of the epidemic is given in red in the inset) and waning phase (blue, true reproduction number $R < 1$ also in blue in the inset) of an epidemic. Solid lines show mean values while the shaded ribbons cover the full range of the information ratios derived from all case curves in (**b**). **b** Shows epidemic trajectories simulated using COVID-19 parameters from renewal models (see Methods) but subject to under-reporting and delays. We plot 1000 case $C_t$ curves, in various colours against the true infections $I_t$ (black). The inset histograms plot differences in crossing times between when the posterior evidence for $R \geq b$ ($\Delta_{\text{grow}}$, red) and $R \leq b$ ($\Delta_{\text{wane}}$, blue), exceed a threshold $\eta$ (see main text), when inferred from $C_1^t$ versus $I_1^t$.

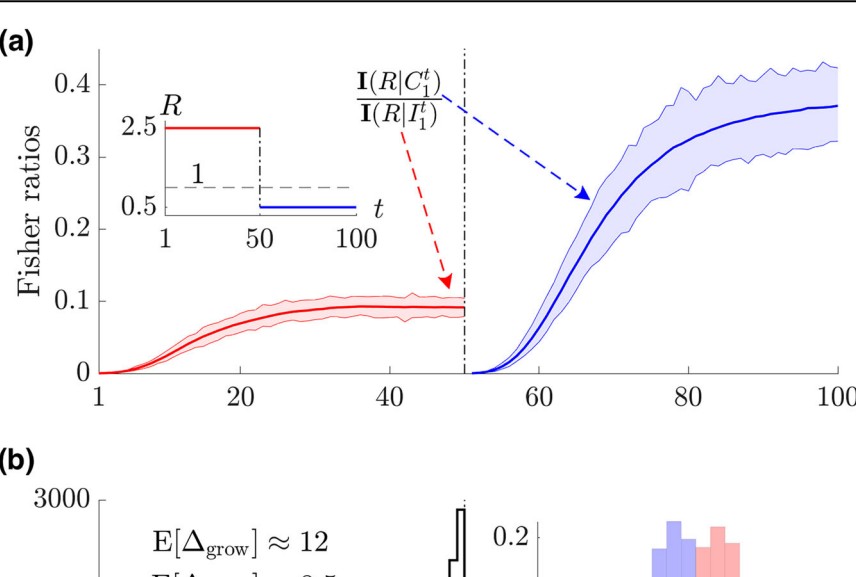

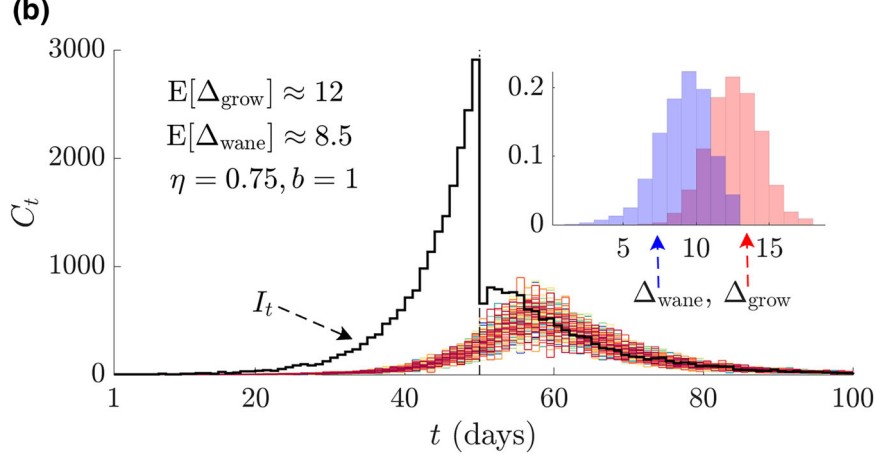

$\mathbf{P}(I_t \leq a|C_1^t) \geq \eta$ for declining ones. We directly use case counts instead of hypothesis probabilities as cases provide the clearest extrinsic signals that may be used to decide if to act or not[3]. Note that we can infer infections from cases and other proxies by simply upsizing the case counts by the noise probabilities (under a popular binomial model)[32,38]. Our central insight is that, irrespective of the values of $\eta$ and $a$, practical noise sources induce an important asymmetry that makes timely interventions in response to resurgences markedly more difficult e.g., in Fig. 1 at every $a$ we find that $\mathbf{E}[\Delta_{\text{grow}}] > 6\mathbf{E}[\Delta_{\text{wane}}]$. Next, we test if these asymmetries hold when intrinsic transmissibility signals are used instead.

**Responding to epidemic growth and decline using transmissibility estimates**

The effective reproduction number, $R$, is a popular metric of infectious disease transmissibility that is frequently used to inform public health policymaking[39]. The value of $R$ is compared to a threshold of 1 to indicate whether the epidemic will grow or wane. Because $R$ is a latent parameter of the epidemic, it needs to be estimated from observed data. The incidence of new infections, $I_t$, contains the most information for inferring $R$, but practical surveillance biases mean that we frequently can only observe new cases, $C_t$. This necessarily results in a loss in information about $R$, which leads to an increase in our estimate uncertainty. We can quantify these losses by computing the Fisher information of $R$ from the incidence of infections $\mathbf{I}(R|I_1^t)$ and cases $\mathbf{I}(R|C_1^t)$[32,40]. The Fisher information is an important measure because it defines the smallest asymptotic uncertainty achievable by any unbiased estimator[27].

We may use estimates of $R$ to inform actions by setting our decision problem according to the posterior evidence $\mathbf{P}(H_1|X_t) \equiv \mathbf{P}(R \geq b|C_1^t)$ and $\mathbf{P}(H_1|X_t) \equiv \mathbf{P}(R \leq b|C_1^t)$ for growing and waning epidemics respectively, with some threshold $b$. As we aim to derive analytic insights about our ability to solve these decision problems generally, we consider asymptotic limits at

which the Fisher information measures the uncertainty around our $R$ estimates. At these limits the posterior distribution of $R$ is Gaussian with mean at its maximum likelihood estimate (which converges to the true $R$) and variance $\sigma^2$ inversely proportional to the Fisher information. The $b$ that we should act on, given $\eta$, relates to the quantile function $\sqrt{2}\sigma\,\text{erf}^{-1}(2\eta - 1)$ (directly for $\mathbf{P}(R \leq b|C_1^t)$ or its complement for $\mathbf{P}(R \geq b|C_1^t)$), with erf as the error function.

Consequently, for any decision cost-threshold $\eta$ the Fisher information plays a central role. We model epidemics using renewal processes as in the Methods and Fig. 1 and assume that cases derive from infections with a reporting proportion of $\rho_s$ at time $s$ and cumulative delay probability $F_{t-s}$ for reports delayed by at most $t - s$ time units. We compute the Fisher information about $R$ from case data $\mathbf{I}(R|C_1^t)$ as on the left of Eq. (4) with $\Lambda_s \overset{\text{def}}{=} \sum_{x=1}^{s-1} w_{s-x}I_s$ as the total infectiousness of the disease and $w_{s-x}$ as the probability of an infection being transmitted in $s - x$ time units. Equation (4) follows from the Methods and the framework in refs. 32,41 and involves summing along the period over which $R$ is (approximately) constant.

$$\mathbf{I}(R|C_1^t) = \frac{1}{R}\sum_{s=1}^{t}\rho_s F_{t-s}\Lambda_s, \quad \frac{\mathbf{I}(R|C_1^t)}{\mathbf{I}(R|I_1^t)} = \frac{\sum_{s=1}^{t}\rho_s F_{t-s}\Lambda_s}{\sum_{s=1}^{t}\Lambda_s} \leq 1. \quad (4)$$

If we set $\rho_s = 1$ for all $s$ (perfect reporting) and $F_0 = 1$ (no delays) then we recover the Fisher information from the infection counts $\mathbf{I}(R|I_1^t)$. This measures the intrinsic uncertainty from the stochasticity of transmission. For example, if infections are few, then $\Lambda_s$ and $\mathbf{I}(R|I_1^t)$ are small, reflecting inherent limitations to inferring spread in these settings. We cannot compute $\mathbf{I}(R|C_1^t)$ without knowledge of the noise sources corrupting surveillance data, but we can extract key insights by taking the ratio of the case to infection values, $\frac{\mathbf{I}(R|C_1^t)}{\mathbf{I}(R|I_1^t)}$, as on the right of Eq. (4). This reveals that the losses

in information about $R$ arising from the noise in the cases depend multiplicatively on noise terms $\rho_s F_{t-s}$ weighted by $\Lambda_s \left( \sum_{s=1}^{t} \Lambda_s \right)^{-1}$ across time.

As delays are always forward in time and induce more information loss towards the present $t$ (i.e., $F_0 = \min F_{t-s}$ as it is cumulative) we find an inherent asymmetry in our ability to make decisions for growing versus waning epidemics. Growing epidemics have an increasing $\Lambda_s$ so $\Lambda_s \left( \sum_{s=1}^{t} \Lambda_s \right)^{-1}$ rises as $s \to t$, contributing more to the sum in Eq. (4). Consequently, the noise from the delay has a magnified effect on the overall estimate uncertainty, limiting our evidence $\mathbf{P}(H_1 | X_t)$. The converse occurs for declining epidemics. In Fig. 2 we plot the Fisher information ratios from the simulations underlying Fig. 1 (which use realistic noise parameters) and expose this performance gap, with growing epidemics having ratios that are approximately 3 times smaller. Note that as we shrink the noise towards perfect reporting (not shown), both our ratios tend to 1 (as expected) but the convergence during the growth phase is slower.

We also consider one example of the practical consequences of this asymmetry in the Fisher information ratios in the inset of Fig. 2. There we apply an estimation method (EpiFilter[42]) to generate posterior distributions $\mathbf{P}(R \geq b | I_1^t)$ and $\mathbf{P}(R \geq b | C_1^t)$ for the growing portion of the epidemic and $\mathbf{P}(R \leq b | I_1^t)$ and $\mathbf{P}(R \leq b | C_1^t)$ for the waning period and record differences in times for when these measures cross a threshold $\eta$. We again find appreciable asymmetry but note that the discrepancies are not as large as in Fig. 1. This is unsurprising because the two approaches are not directly comparable and the posterior estimates of $R$ also depend on factors such as the assumed prior distributions, generation time distribution choices and the smoothing method used. This is why we examined the Fisher information. Nevertheless, the asymmetry is apparent.

The main alternative to $R$ is the epidemic growth rate $r$, which has similar properties[39,43] and indicates resurgence or control based on whether it is positive or negative. The growth rate is sometimes preferred to $R$ because $r$ offers a measure of the speed or timing needed from an intervention[44] and are more robust to generation time distribution misspecifications. These benefits come at the expense of some loss in mechanistic interpretability and in making smoothing assumptions that may induce other biases[43]. We can broadly relate $R$ and $r$ for a given generation time distribution via the Euler-Lotka equation[36]. We make the common assumption of a gamma generation time distribution with shape $\alpha$ and scale $\beta^{-1}$ parameters and apply Eq. (10) of the Methods to derive the Fisher information on the left of Eq. (5).

$$\mathbf{I}(r | C_1^t) = \frac{\alpha}{\beta} \left( 1 + \frac{r}{\beta} \right)^{\frac{\alpha-2}{2}} \sum_{s=1}^{t} \rho_s F_{t-s} \Lambda_s, \quad \frac{\mathbf{I}(r | C_1^t)}{\mathbf{I}(r | I_1^t)} = \frac{\mathbf{I}(R | C_1^t)}{\mathbf{I}(R | I_1^t)}. \quad (5)$$

Interestingly, we observe that the noise terms appear in the same way as Eq. (4) and that the remaining terms are independent of the noise. As a result, when we take the ratio of case to infection data Fisher information values, as in the right of Eq. (5), we obtain precisely the same result as in Eq. (4), despite the Fisher information itself being different for $r$. The noise induced asymmetry observed for $R$ therefore also affects $r$ and likely cannot be overcome by changing our measure of epidemic transmissibility. We confirm this further by noting that recent metrics from[45] that reformulate $\Lambda_s$ to reduce its dependence on generation times have similar Fisher information formulae that will also show this skew. Consequently, optimal decision thresholds for growing epidemics are substantially more difficult to resolve than equivalent thresholds for waning epidemics and are unlikely to improve by changing how policymakers track spread.

## Asymmetry persists under complex (feedback) decision frameworks

In previous sections we demonstrated that surveillance noise induces asymmetric limits to optimal decision-making when comparing growing and waning epidemics. Since we aimed to extract generalisable insights, the models we investigated, while commonly used to study real epidemics[39], were somewhat simplified. Here we confirm that the asymmetry we uncovered remains an intrinsic barrier to performance even when cost-

optimal decision frameworks that leverage feedback control theory[46,47] and Bayesian optimisation are employed. Using the same parameters from Fig. 2 (see Methods), we simulated COVID-19 epidemics and applied the model predictive control algorithm from[28] to optimise the timing of decisions. This algorithm aligns with the framework of Eq. (1) but explicitly evaluates and propagates the costs of both action and inaction at every decision time (weekly).

We detail these costs in Eq. (11) of the Methods, including terms that account for the disruption (both economic and other types) of initiating an NPI and penalties for the trajectory of the epidemic (e.g., peak and endemic infection loads induce costs linked to healthcare burdens) expected to occur due to our NPI decisions. We model a lockdown as a multiplicative reduction in $R$, using parameters consistent with[48]. We find the optimal time for initiating (relaxing) a lockdown for a growing (waning) epidemic, by minimising Eq. (11) across a projected horizon that considers longer-term feedback or rebound effects resulting from action or inaction. This procedure uses case thresholds, considers uncertainties in $R$, which are inferred as part of the algorithm and applies explicit costs to actions, uniting all the previous Results sections.

We present results of this algorithm in Fig. 3 for simulated COVID-19 epidemics. We compute optimal decision times for the perfect (noiseless) case and for scenarios featuring realistic surveillance noise (under-reporting and delays in line with previous sections). Analysing the start, end and length of lockdowns (histograms in Fig. 3), we observe that noisy surveillance causes lags of 2-4 weeks when initiating an NPI. However, this same noise induces delays of only under 1 week when relaxing that NPI. The growth-waning asymmetry is well-defined and persistent even under sophisticated decision algorithms. In Fig. 4 we show equivalent analyses for epidemics simulated under Ebola virus parameters (see Methods for details), verifying the consistency of our claims. Noise causes delays of 4-5 weeks when initiating lockdowns but only 1–2 weeks when relaxing them. Absolute times are longer for Ebola virus due to its slower timescale of spread (its mean generation time[49] is more than double that of COVID-19). As a result, the bottleneck we have discovered is resistant to sophisticated predictive algorithms and suggests that proactive policymaking action is necessary during epidemic growth.

## Discussion

Policymakers often have to make critical public health decisions from uncertain and unreliable data. While research has shown how this uncertainty can markedly impact (often in unintuitive ways) the timing, selection and success of interventions[7,12,16,18,28], much is still unknown about how real-time decision-making can best incorporate or mitigate this uncertainty. Recent studies have proposed integrating uncertainty within rigorous decision theory frameworks[50–52]. Here, we support this proposal, but instead of applying complex modelling to guide specific action or inaction, we concentrated on extracting generalisable insights about how uncertainty fundamentally limits decision-making. We tackled this problem by exploring how predominant forms of surveillance uncertainty or noise, relating to under-reporting and delays in reporting cases, present bottlenecks for real-time decisions based on two of the most common outbreak indicators – the incidence of cases and the reproduction number (or related growth rate).

We discovered that there is a surprising, intrinsic and important asymmetry induced by these sources of noise that results in substantially reduced performance of the solutions to optimal decision problems for growing epidemics, relative to the performance on equivalent problems for waning epidemics. This asymmetry remains irrespective of whether we base decisions on extrinsic proxies of infection incidence (Fig. 1) or intrinsic estimates of transmissibility (Fig. 2). We found theoretical justification for this asymmetry using information and Bayesian decision theory and then confirmed this asymmetry remained when complex algorithms were used to explicitly optimise decisions according to the costs of action as well as the opportunity costs of inaction (Figs. 3, and 4). While surveillance noise is expected to restrict actionable

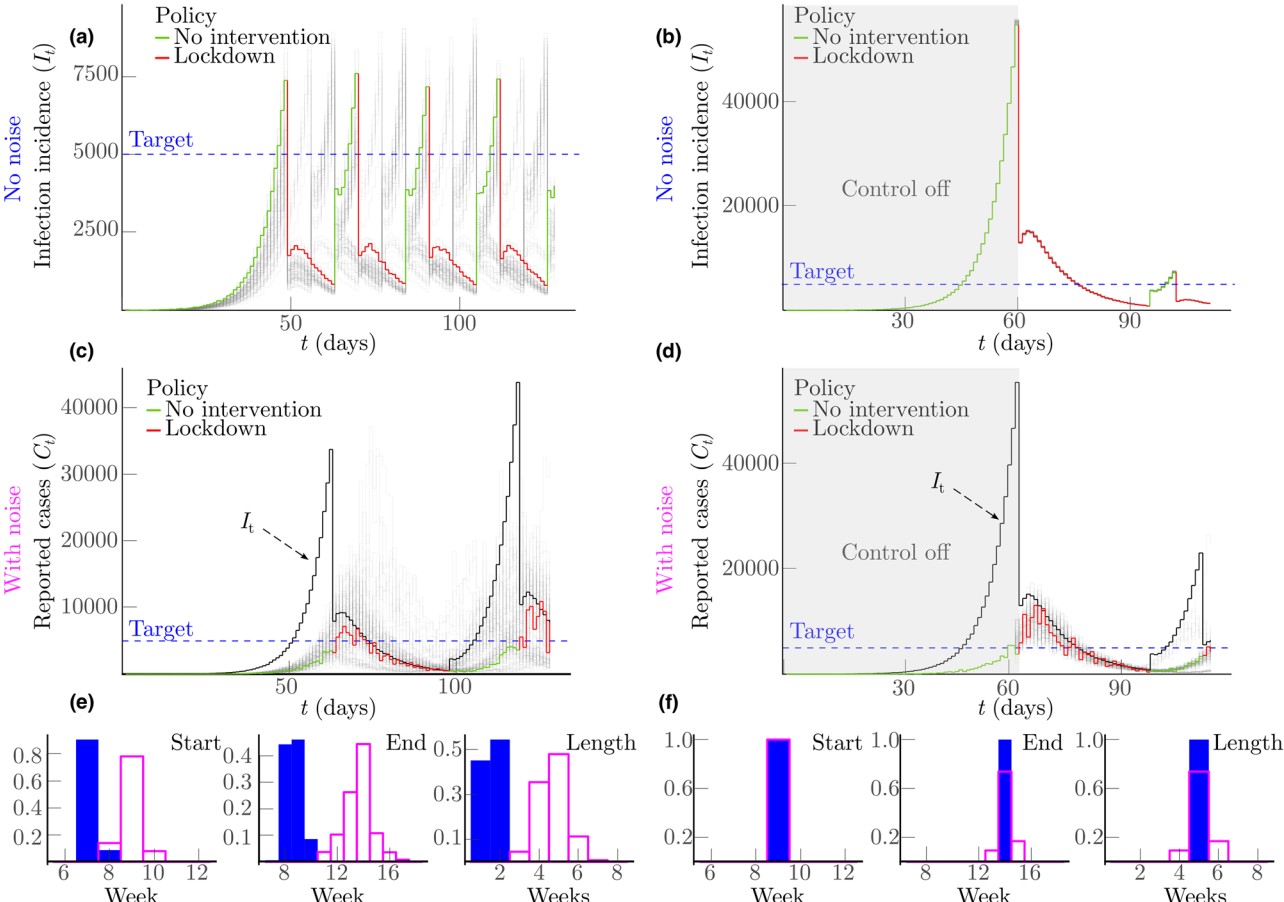

**Fig. 3 | Explicitly optimised NPI initiation and relaxation under noisy surveillance. a, b** Present simulated epidemics under COVID-19 parameters (see Methods) and renewal models, with interventions implemented by the model predictive control (MPC) algorithm from[28]. This projects the costs of action or inaction forward in time together with the outcomes of those choices (e.g., larger epidemic peaks or endemic infection loads) and optimises timing based on minimising costs over the projections (see Methods). We plot multiple stochastic epidemic realisations in light shades with one trajectory highlighted. There is no noise, so we observe the true infections. **a** Focuses on growing epidemics and the optimal initiation of the first lockdown. In some cases, multiple lockdown actions are visible because the MPC algorithm continually optimises costs, but we focus only on the statistics of the first decision (in line with Figs. 1–2). **b** Examines scenarios where no interventions were made for the first nine weeks after which the MPC algorithm initiated a lockdown and focuses on the waning components of this epidemic and optimally releasing that

lockdown. **c, d** Repeat the simulations of **a, b**, but now we cannot access the true infections (shown in black) and must decide optimal actions based on case curves subject to surveillance noise (delays and under-reporting). Again, multiple stochastic epidemic realisations are in light shades with one trajectory highlighted. **e, f** Plots histograms of the times for lockdown start, end and duration for the perfect (shaded) and noisy (unshaded) scenarios. **e** Focuses on differences in the control problems of **a** and **c**, while **f** considers differences from **b** and **d**. Note that because for waning epidemics the beginning of the lockdown is the same both with and without noise, outcomes only deviate afterwards. The difference in the corresponding histograms shows the impact of the surveillance noise. Our results indicate that optimally relaxing a lockdown or another related NPI is minimally affected by imperfect surveillance. The related histograms also show smaller discordance (the shaded and unshaded ones are closer together).

information, it is not obvious that the bottleneck it imposes should be notably more restrictive during growth.

There are several important ramifications of this asymmetry. First, as this innate performance restriction on responding to epidemic growth is likely further compounded by scarcer data and additional unknowns (e.g., transmissibility can be poorly specified during epidemic emergence[12]), proactive interventions may be necessary to achieve timely control[7,23]. In contrast, a more reactive approach to relaxing interventions is likely sufficient. Proactive policymaking may require reducing the threshold for action (potentially elevating the likelihood of false alarms) in response to routine data or leveraging other data streams to extract early-warning signals of resurgence (potentially increasing disease monitoring costs or infrastructure requirements) that can guide timelier decisions. Sentinel surveillance of species that may be sources of zoonotic spillovers or digital monitoring of online search and sentiments may offer such signals[53,54].

Second, if surveillance noise can be reduced by reallocating resources (e.g., rapid diagnostic tests, data sharing agreements) or relying on alternate data (e.g., deaths and hospitalisations may be more reliable than cases early

on, though potentially more delayed) then deployment should prioritise growing stages of outbreaks. More localised surveillance (e.g., gathering data at community versus regional levels or by age group[23,55]) may also help mitigate inherent asymmetry in detecting resurgence because emerging infections appear first at small scales (though data are scarcer at these scales) before the epidemic becomes widely established. Third, when NPI effectiveness is retrospectively assessed it may be valuable to contextualise performance against the limits imposed by the bottlenecks we have explored to gain a more objective evaluation. This provides a clearer reference for effectiveness, ensuring we are not comparing against unrealistic ideals or infeasible counterfactual scenarios.

Although the asymmetry we uncovered is fundamental to noisy epidemic surveillance, there are limitations to our analyses. We only considered homogeneous (well-mixed) transmission models. The impact of realistic heterogeneities in spread due to geography, demography and other characteristics may influence what an optimal decision should be and hence the level of asymmetry. This is an open question that we aim to explore in the future. Additionally, we have not examined how emerging epidemic data

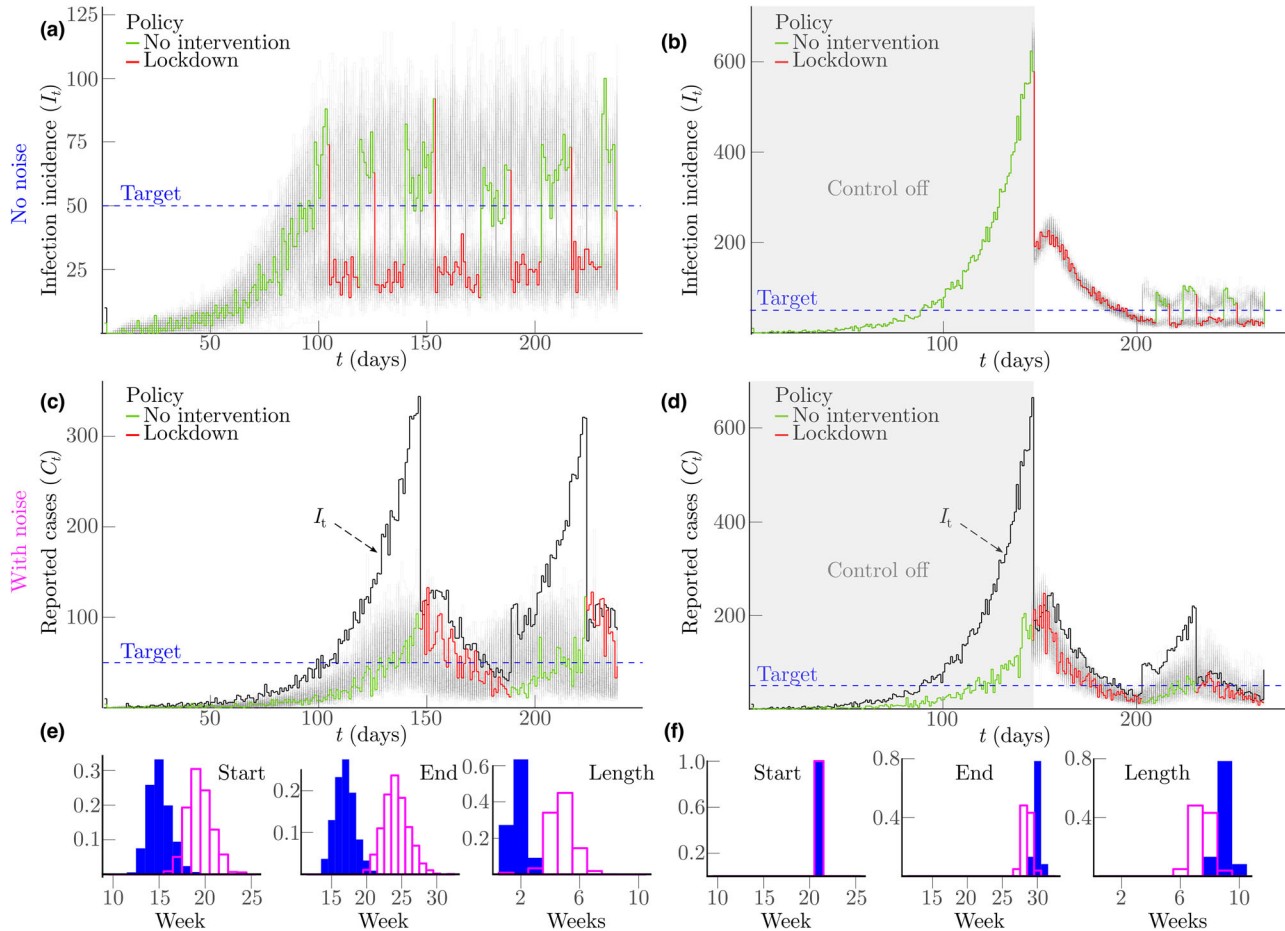

**Fig. 4 | Persistent noise-induced asymmetry under optimal NPI initiation and relaxation. a–f** Repeat the simulations and analyses of Fig. 3 but under Ebola virus transmission and surveillance noise parameters (see "Methods" for details). **a** and **c** Provide the optimal control performance for initiating a lockdown as epidemics grow under no noise and practical surveillance respectively. **e** Shows the histograms of lockdown start, end, and duration times for the perfect (shaded) and noisy (unshaded) scenarios. **b** and **d** Present the optimal MPC performance for lifting a lockdown as epidemics wane under no noise and practical surveillance, respectively. **f** Plots corresponding histograms of timing and duration performance. We find the asymmetry in Fig. 3 therefore, persists.

could help us overcome these bottlenecks. Recent initiatives[56] have proposed integrating wastewater surveys and genomic sequencing within standard surveillance schemes. While promising, the potential of these data for enhancing early warnings of outbreaks is still being assessed. Should these data feature some latency and under-ascertainment[32], we expect decision-making asymmetries to re-emerge.

Lastly, we constrained our analyses to binary decisions between action and inaction based on data available in real time. This was both to embody the common situation where policymakers must make choices from the latest (updating) data and to allow for analytic and generalisable insights to be extracted. However, this framework neglects that optimal decisions could involve more complex trade-offs among multiple, simultaneous NPI options with differing benefits and costs informed by both real-time and historical data from other outbreaks or past experience with those NPIs. Although some complex decision problems can be reduced to generalisations of the binary case we study, yielding multiple decision thresholds[26,57], it remains unknown if and when these complexities could modify the performance asymmetry that we uncovered, particularly when combined with the heterogeneities and auxiliary data above.

These limitations notwithstanding, the consistent asymmetry we found, using both elementary mathematical arguments and sophisticated (predictive) algorithms, underscores a meaningful directionality in how surveillance imperfections shape and bound decision-making in the face of uncertainty. Knowing these asymmetric performance bounds is not only helpful for deciding when interventions need to be proactive or reactive, but

also for diagnosing how our responses to outbreaks may fail and for justifying why faster and more stringent measures are necessary. With mounting calls for better integration of formal decision frameworks[16,18,47,50] together with enhanced and multimodal surveillance data[37,56], epidemic models will only become more complex and difficult to interrogate. Having generalisable insights into performance limits can help anticipate failures in surveillance-driven decisions and improve their robustness.

## Methods
### Bayesian decisions from epidemiological data
Bayesian decision theory offers a formalised and rigorous way of informing decisions with data and under uncertainty[26,27]. Here, we rework some classical results to gain insight into how we can optimise intervention decisions. We examine a binary decision or hypothesis testing problem where, given information $X_t$ on the dynamics of an infectious disease up to time $t$ (e.g., cases or estimates of transmissibility), we want to decide to act now (at time $t$) or to wait for more information to accrue. We let $H_0$ be our null hypothesis, which defines the situation that we should do nothing. $H_1$ is the alternative hypothesis that we should act now.

These hypotheses are reassessed sequentially with time and delineate optimal decision points for action. While we consider binary decision-making only, the expressions below are known to generalise to complex decision problems involving multiple hypotheses. Solutions to those problems are qualitatively similar[57]. We define $c_{ij}$ as the cost of acting according to $H_i$ when $H_j$ is true. A FP action (or false alarm) occurs when we act before

we should and has cost $c_{FP} = c_{10}$. A FN or missed action has cost $c_{FN} = c_{01}$. TP and TN, i.e., correct actions have respective costs of $c_{TP} = c_{11}$ or $c_{TN} = c_{00}$. The expected cost of a decision $i$, $\mathbf{E}[c_i]$, and the associated optimal action $i_t^* = \min_{i \in \{0,1\}} \mathbf{E}[c_i]$, follow as in Eq. (6). There $\mathbf{P}(H_j|X_t)$ is the posterior distribution of the evidence for hypothesis $j \in \{0, 1\}$ given available information $X_t$[57].

$$\mathbf{E}[c_i] = \sum_{j \in \{0,1\}} c_{ij} \mathbf{P}(H_j|X_t), \quad i_t^* = \mathbf{1}\left(\frac{\mathbf{P}(H_1|X_t)}{\mathbf{P}(H_0|X_t)} \geq \epsilon = \frac{c_{FP} - c_{TN}}{c_{FN} - c_{TP}}\right). \tag{6}$$

Equation (6) states that the optimal action depends on a threshold or decision boundary $\epsilon$ that solely depends on the costs. This standard result from decision theory is already interesting as even when we do not know these costs accurately, we can still assess their likely effects via different choices of $\epsilon$. Moreover, if we include prior distributions on each hypothesis, Eq. (1) defines a likelihood ratio test. Equation (6) also generalises to multiple hypotheses[29], where we would find different thresholds for every hypothesis. We can rearrange our decision rule from Eq. (6) by recognising that $\mathbf{P}(H_0|X_t) = 1 - \mathbf{P}(H_1|X_t)$ to obtain Eq. (1) of the main text, with $\eta = \frac{\epsilon}{1+\epsilon}$.

We can also apply information theory to derive insights into this decision problem. We wish to communicate one bit of information to a decision maker, i.e., a 1 or 0 to indicate that the evidence indicates that action or inaction, respectively, is required. If $H_1$ is true, then the evidence for action $\mathbf{P}(H_1|X_t) = p_t$ will accumulate. The Shannon entropy of a Bernoulli process with success probability $p_t$ is $\mathcal{H}_{p_t} = p_t \log_2 \frac{1}{p_t} + (1 - p_t) \log_2 \frac{1}{1-p_t}$ and defines the uncertainty for a distribution over two choices with probabilities $p_t$ and $1 - p_t$[40]. Logarithms are to the base 2 so $0 \leq \mathcal{H}_{p_t} \leq 1$ is in bits. In Eq. (7) we take the negative derivative of this function with $p_t$, showing how this uncertainty reduces as we accrue evidence to act.

$$\frac{d\mathcal{H}_{p_t}}{dp_t} = \log_2\left(\frac{1 - \mathbf{P}(H_1|X_t)}{\mathbf{P}(H_1|X_t)}\right), \quad i_t^* = \mathbf{1}\left(-\frac{d\mathcal{H}_{p_t}}{dp_t} \geq \log_2 \epsilon\right). \tag{7}$$

Consequently, optimal decision-making compares the loss of randomness or rise in certainty that $H_1$ is true with the logarithm of the cost-based threshold from Eq. (6). This is characterised by $-\frac{d\mathcal{H}_{p_t}}{dp_t}$, which is a logit function. The speed at which we cross our decision threshold, i.e., the cost-adjusted amount of time $t^*$ that we should wait before acting, is controlled by how quickly $p_t$ grows with time to surpass our evidence-based threshold[30].

### Renewal models with practical surveillance

The renewal branching process[58] is widely used to model infectious disease epidemics of COVID-19, Ebola virus, pandemic influenza and many others[59]. It simulates the incidence of new infections $I_t$ at some time $t$ in terms of the effective reproduction number $R$ and past incidence $I_t^{t-1} \equiv \{I_1, I_2, \ldots, I_{t-1}\}$ as in the left of Eq. (8) with **Pois** as Poisson noise. Here, $\Lambda_t$ is the total infectiousness and encodes the impact of past incidence, i.e., $\Lambda_t \overset{\text{def}}{=} \sum_{x=1}^{t-1} w_{t-x} I_x$. The $w_{t-x}$ terms are the probability of an infection being transmitted in $t - x$ time units and define the generation time distribution of the disease with $\sum_{u=1}^{\infty} w_u = 1$[36].

$$I_t \sim \mathbf{Pois}(\Lambda_t R), \quad C_t \sim \mathbf{Pois}\left(R \sum_{s=1}^{t} \rho_s \delta_{t-s} \Lambda_s\right). \tag{8}$$

Normally, infections are not observable, so the standard renewal model is modified to describe the incidence of symptomatic cases, $C_t$ at time $t$, instead. This requires including noise terms such as reporting fractions, $\rho_s$ at time $s$, and the probability, $\delta_{t-s}$ of a reporting delay of $t - s$ time units. This

leads to the right side of Eq. (8)[32,38]. In all of these equations, we assume a constant $R$ until time $t$, but note that they remain valid for time-varying reproduction numbers (in which case $R$ becomes an approximate mean). If we have no delay and perfect reporting then $\rho_s = 1$, $\delta_0 = 1$, $\delta_{x>0} = 0$ and hence $C_t = I_t$, recovering the original renewal model.

In Figs. 1–3, we simulate under established COVID-19 transmission and surveillance noise parameters from the literature. We use the generation time distribution from ref. 10 (a gamma distribution with a mean of 6.5 days) and model delays in reporting, as done in ref. 60 (a negative binomial distribution with a mean of 10.8 days) with the fraction of infections reported as cases based on ref. 61 (a beta distribution with mean of 0.38, implying under-reporting of mean 0.72). In Fig. 4 we perform equivalent simulations to Fig. 3 but now under Ebola virus parameters. We apply the generation time distribution from ref. 49 (a gamma distribution with a mean of 15.3 days) and reporting noise from[62,63] (delays follow a negative binomial distribution with a mean of 11.8 days and under-reporting conforms to a beta distribution with a mean of 0.4). For the precise distributions underlying both our COVID-19 and Ebola virus analyses (as well as code to reproduce these figures) see the Data and code availability section.

### Fisher information given noisy observations

The minimum (asymptotic) uncertainty around estimates of $R$ derived from the renewal model of Eq. (8) can be quantified using the Fisher information $\mathbf{I}(.)$. This can be computed from the expected curvature of the log-likelihood of the statistical models from Eq. (8). This leads to the relations in Eq. (9), which provide the information from infections and cases respectively, and are adapted from the frameworks introduced in refs. 32,41. Here $F_{t-s} = \sum_{x=0}^{t-s} \delta_x$ describes the cumulative reporting delay probability and $\mathbf{I}(R|I_1^t) \geq \mathbf{I}(R|C_1^t)$ (noise reduces information).

$$\mathbf{I}(R|I_1^t) = \frac{1}{R} \sum_{s=1}^{t} \Lambda_s, \quad \mathbf{I}(R|C_1^t) = \frac{1}{R} \sum_{s=1}^{t} \rho_s F_{t-s} \Lambda_s. \tag{9}$$

In the main text, we examine ratios of these information terms and also derive related ratios for epidemic growth rates $r$. These growth rates have a mapping to $R$ described by the Euler-Lotka equation[36]. This equation depends on the generation time distribution of the disease. Commonly, a gamma-distributed generation time distribution is assumed[43,59] i.e., the $w_u$ probabilities describe a shape-scale $\mathbf{Gam}(\alpha, \beta^{-1})$ distribution. Under this setting the Euler-Lotka relationship $f(.)$ takes the form of the left side of Eq. (10).

$$R = f(r) = \left(1 + \frac{r}{\beta}\right)^{\alpha}, \quad \mathbf{I}(r|X_1^t) = \mathbf{I}(f(r)|X_1^t)\left(\frac{df}{dr}\right)^2. \tag{10}$$

For a given generation time distribution, $R$ is a smooth function of $r$, $f(r)$, and we may apply the Fisher information change of variables formula on the right side of Eq. (10) to convert the expressions for $R$ in Eq. (9) to ones for the growth rate. Here $X_1^t$ may be the time series of cases $C_1^t$ or infections $I_1^t$ as required. In the main text, we use the above equations to derive Fisher information ratios for both reproduction numbers and growth rates.

### Cost-optimal feedback control of epidemics

We have focused on extracting general insights about the asymmetries in response times to growing or waning epidemics under arbitrary cost thresholds. This required tractable modelling approaches that facilitate analytic results. However, it is possible to instead construct complex algorithms[15] that directly integrate competing costs from interventions and disease burden (e.g., epidemic peaks and infection endemic levels), as well as consider longer-term, feedback effects from chosen interventions[47]. We adopt such an algorithm from[28], which uses model predictive control

(MPC)[46] to balance the component costs of $\psi(t)$ in Eq. (11).

$$\psi(t) = \alpha\left|I_t - I_{\text{end}}\right| + \beta\mathbf{1}\left(I_t > I_{\text{pk}}\right) + \phi(\text{NPI}_t). \qquad (11)$$

This MPC algorithm considers costs due to (i) the difference between infections and a practical endemic goal $I_{\text{end}}$ (weighted by $\alpha$), (ii) exceeding a peak value $I_{\text{pk}}$, which for example models the level of infections beyond which healthcare resources are overrun (weighted by $\beta$) and (iii) the actual economic or other penalties from enforcing an NPI (wrapped in a flexible function $\phi(.)$). Heuristically, applying a more stringent NPI increases (iii) but can decrease (i) and (ii). Hence, we penalise both action and inaction. This aligns and expands on the framework from Eq. (1) and implies a threshold for action based on how components (i)-(iii) balance. The MPC approach of ref. 28 finds the optimal time to initiate or relax an NPI by minimising the long-term costs of those actions over a horizon $h$ i.e., $\sum_{s=0}^{h}\gamma^s\psi(t+s)$, with $\gamma$ as a discount factor.

This is done by projecting the consequences of those choices using a generalised form of Eq. (8) and then applying Bayesian optimisation to select the cost-optimal choice. In computing projections, the algorithm infers the effective reproduction number, incorporating uncertainty from transmissibility. See ref. 28 for complete details of this procedure. In our analyses $\text{NPI}_t$ is 0 when inactive at time $t$. When active it is 1 and models a lockdown or stay-at-home order as a multiplicative reduction in $R$ (though it can be easily modified to model other intervention types). The cost $\phi(1)$ associated with this action is set based on analyses in ref. 48. Late NPI relaxation increases (iii), while late NPI initiation increases (i)-(ii). When surveillance is noisy, we replace $I_t$ with cases $C_t$ and apply appropriate delay and under-reporting parameters from the literature. Projections using $I_t$ consider the stochasticity of transmission, while those using $C_t$ factor in the additional noise from the surveillance imperfections.

## Data availability
All relevant data are available from the authors upon request.

## Code availability
All code to reproduce the analyses and figures of this work are freely available (MATLAB and R) at: https://github.com/kpzoo/asymmetricDetection with release https://doi.org/10.5281/zenodo.17184842.

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

## Acknowledgements

KVP and SB acknowledge support (Reference No. MR/X020258/1) from the MRC Centre for Global Infectious Disease Analysis funded by the UK Medical Research Council. This UK-funded grant is carried out in the frame of the Global Health EDCTP3 Joint Undertaking. The funders played no role in study design, data collection and analysis, decision to publish, or manuscript preparation.

## Author contributions

Conceptualization, investigation, formal analysis, writing (original draft preparation), funding acquisition, supervision: K.V.P. Software, visualisation: KVP and SB. Validation, methodology, writing (review and editing): K.V.P., B.L., C.A.D., S.B.

## Competing interests

The authors declare no competing interests.
