## [Transparent Peer Review file · Communications Physics]

Asymmetric limits on timely interventions from noisy epidemic data

Corresponding Author: Dr Kris Parag

Version 0:

Reviewer comments:

Reviewer #1

(Remarks to the Author)

In this manuscript, the authors demonstrate the potential costs of missed actions or false alarms in outbreak responses through simulations. This study provides a critical and insightful perspective on the asymmetric impact of surveillance noises on optimal decision-making, depending on whether an epidemic is in a growing or waning phase. While the manuscript is well-written, certain aspects could be further clarified to enhance its impact. I hope my suggestions contribute to making this work even stronger.

1. While the authors explain their findings well mathematically in the Results section, I suggest interpreting the results in a more applicable way (focusing more on practical interpretations of findings) for public health decision-makers, who are the primary users of these findings.
2. Case reports in the very early phase of an epidemic can be highly uncertain, particularly when dealing with a newly emerging pathogen (like COVID-19 in February). I believe, in this context, hospitalization and death counts may serve as more reliable indicators in the early phase of the epidemic, despite their longer reporting delays (which is likely to make an asymmetric impact of surveillance noises more significant).
4. When case counts are below 100, the exponential growth rate (or effective reproduction number) is often highly uncertain in practice, fluctuating significantly due to stochasticity. This introduces an additional source of noise in both exponential growth rates and reproduction number estimates, separate from reporting delays and under-reporting. I would be interested to know whether this factor is accounted for in the current analyses.
5. The legend for Figure 2 could be more detailed to improve clarity. For example, it would be helpful to define the terms "supercritical R" and "subcritical R" and clarify the corresponding y-axis labels. Additionally, I found it difficult to interpret the various ribbon colors overlaying newly infected cases based solely on the current legend. A more detailed explanation would help clarify their meaning. The same comment applies to Figures 3 and 4 to ensure that their messages are conveyed.
6. I believe the exponential growth rate (which was used in the method with case counts) and reproduction numbers could be considered somewhat interchangeable (especially in the early phase when we can assume there is negligible population-level immunity). Given this, I am curious about the rationale behind treating these two concepts differently in the analysis. Could the authors clarify the distinction they made and why they chose to approach them separately?
7. It would be beneficial if the authors discussed potential strategies or directions for further studies that could address the asymmetry issues highlighted in the study. This would provide readers with a broader perspective, not only on how outbreak responses may fail but also on ways to mitigate such failures.

[Minor comment]

1. I suggest that the authors explicitly define the reproduction number in the Results section. In this context, the term "effective reproduction number" may be the right choice?

Reviewer #2

(Remarks to the Author)
Report Communications Physics

This study reveals that noisy surveillance data, affected by reporting delays or case underreporting, hampers asymmetrically the optimal management of epidemics. Namely, during epidemic growth, both noise sources contribute to the accumulation of delays for the timely deployment of control strategies whereas, in the epidemic declining phase, underreporting compensates the lag in case reports and allows for a more proper lifting of interventions once the outbreak is controlled.

The manuscript is well-written and, to the best of my knowledge, the asymmetric contribution of noise sources for the timely intervention on epidemics represents a novel finding for the field of mathematical epidemiology. In addition, I find the study very timely and the results reported here are relevant, as addressing the inherent constraints limiting the design and implementation of control strategies improves our preparedness and response to epidemic scenarios. The manuscript is sound and combines results from simulations, using different frameworks, with some analytical insights obtained from a simplistic mean-field model. Because of all these factors, in my opinion, the study has potential to be published in Communications Physics. Nevertheless, I have some major comments on the manuscript that I would like the authors to address:

- The manuscript should be restructured to be focused on the main findings derived from the study. Namely, the manuscript starts with an exhaustive description of Bayesian decision theory. This section is informative and useful to become familiar with a probabilistic grounded decision theory. However, most of the contents explained there are not essential for the reader to understand the results reported in the manuscript. For this reason, I would encourage the authors to move this section as a subsection of the Methods section.

- The authors correctly identify the origin of the asymmetry but the explanation of the compensation of both noise sources is not strictly correct. I agree that Eq. (4) holds for the exponential growth of epidemics, being r_0 the growth rate. In contrast, the declining phase of epidemics does not always have to be strictly exponential with a fixed exponent, especially when $R \simeq 1$. If exponential, the decay rate is shaped by R rather than by r_0 and should incorporate the fraction of susceptible population. Therefore, it is not reflected by the initial growth rate r_0 , as suggested in Eq (5). To provide a more rigorous explanation, I would recommend the authors to frame the explanation with general rates r for the exponential growth and decay of epidemics.

- The section on the use of transmissibility as an indicator to enforce/lift interventions interrupts the natural flow of the manuscript. While Figures 1, 3, 4 are generated using simple/complex criteria accounting for case numbers, Figure 2 reflects what happens with the Fisher information for an epidemic in the growth/declining phase. To introduce this figure, the authors motivate the fact that sometimes the indicators used for policy-making is the effective reproductive number. While using this criterion is useful for intermittent policies (producing oscillatory dynamics), it is not useful for the first epidemic wave as the effective reproduction number can only decrease over time (policies should always enforced from the beginning). For this reason, I only find this section useful once the first epidemic curves with epidemic dynamics have been shown. Moreover, current Figure 2 does not illustrate the practical implications of the asymmetry found in the previous section beyond stressing the fact the Fisher information values from the exact and the noisy models are closer during the epidemic declining phase. To solve both issues, I will move the section using the reproduction number to the end of the manuscript and complement the results shown in Figure 2 showing another panel with a histogram of the delay in enforcing/lifting the control policies using the effective reproduction number R as the information for decision-making. In that sense, it would also be interesting to check whether underreporting also counteracts lagged case reports when using R .

- Resolution of Figures 3 and 4 should be improved.

Minor details:

- Use another color different from the tone chosen for the 'green' histograms in Figures 3 and 4 as it is difficult to identify the color. Also use thicker lines for the histogram to be more visible.
- λ_s should be defined in the main text.
- I think that colors of the histograms corresponding to underreporting and delayed reports are inter-exchanged in Figure 1.

Version 1:

Reviewer comments:

Reviewer #1

(Remarks to the Author)

I appreciate the authors' efforts in revising the manuscript in line with the previously provided comments. Overall, I find the manuscript to be thoroughly revised (particularly with the additional clarification regarding Fisher information terms) and would support its publication as it is.

Reviewer #2

(Remarks to the Author)

The authors have addressed most of my concerns and the revised version of the manuscript is much clearer and present a more coherent flow that improves its readability. While I am overall satisfied with the changes introduced, I think that the section "Responding to epidemic growth and decline using reported incidence" still needs to be rewritten to ease the understanding of Equations 2 and 3. Before including any mathematical expression, I would recommend the authors to explicitly mention that cases suffer from underreporting and delay in the reports and define the two mathematical parameters used to account for both factors.

Likewise, the authors should explicitly state that the rates r included in Eq.(3) are different, as in the growth phase corresponds to r_0 while in the declining phase corresponds to the time average of the growth rate over the window of interest (as now noted in the manuscript). Moreover, the comment on τ_p before Eq.(3) hinders understanding the fact that this delay is not an input parameter but arises from the equations as a result of underreporting. To avoid confusion, I would omit that mention and define τ_p explicitly in Equation 3 rather than ρ , as the latter is an input parameter to model underreporting.

Responses to Reviewers: COMMSPHYS-25-0151

We thank the reviewers for their constructive and insightful comments. We list our responses to each point and corresponding changes to the main text below. We also include both clean and tracked-change versions of the new text to highlight our revisions. References made here are listed in a bibliography at the end of this document.

Reviewer #1:

In this manuscript, the authors demonstrate the potential costs of missed actions or false alarms in outbreak responses through simulations. This study provides a critical and insightful perspective on the asymmetric impact of surveillance noises on optimal decision-making, depending on whether an epidemic is in a growing or waning phase. While the manuscript is well-written, certain aspects could be further clarified to enhance its impact. I hope my suggestions contribute to making this work even stronger.

Thanks for the appraisal and helpful suggestions, which have bolstered the paper.

1. While the authors explain their findings well mathematically in the Results section, I suggest interpreting the results in a more applicable way (focusing more on practical interpretations of findings) for public health decision-makers, who are the primary users of these findings.

We agree that making the results accessible to public health audiences is important. We have therefore now inserted clarifying statements around key results, indicating their importance for policymaking and have emphasised the implications of our findings in expanded paragraphs of the Discussion. Here we also provide some likely practical resolutions for overcoming the bottlenecks we discovered (e.g., outlining how multimodal data streams or proactive decision frameworks can help). Additionally, we have (see responses to points 1 and 3 of Reviewer 2) restructured the manuscript, making links between sections clearer and porting less relevant formulae and theory to the Methods to better highlight the public health ramifications of our findings and increase the accessibility for broader readership. However, we do maintain some technical content as we believe that epidemiologists using transmission models to provide evidence for informing health decisions are also a key target audience for this work.

2. Case reports in the very early phase of an epidemic can be highly uncertain, particularly when dealing with a newly emerging pathogen (like COVID-19 in February). I believe, in this context, hospitalization and death counts may serve as more reliable indicators in the early phase of the epidemic, despite their longer reporting delays (which is likely to make an asymmetric impact of surveillance noises more significant).

Thanks for highlighting this useful point. We agree that cases may potentially be less reliable than deaths and hospitalisations during emergent epidemic stages but also highlight that few studies have formally compared these different data for decision-making. One such study (by the authors) found many scenarios in which COVID-19 deaths are less informative than cases despite fluctuations in case ascertainment [1]. However, the reviewer is correct that if deaths and hospitalisations are more reliable indicators (and often they can be), they are still subject to under-reporting and delay processes that could lead to strong asymmetries, as in the paper.

The level of noise and asymmetry will vary with geographical and demographic features (e.g., reporting of COVID-19 mortalities was highly heterogeneous with some countries substantially undercounting deaths [2]) as well as with how such data are leveraged to inform decisions. We have amended the Discussion to broadly underscore the use and limitations of auxiliary data, including comments about deaths, hospitalisation and other sources at different scales. Our central conclusion remains valid – during periods of growth, proactive measures may be justified by the asymmetry induced by noise.

3. When case counts are below 100, the exponential growth rate (or effective reproduction number) is often highly uncertain in practice, fluctuating significantly due to stochasticity. This introduces an additional source of noise in both exponential growth rates and reproduction number estimates, separate from reporting delays and under-reporting. I would be interested to know whether this factor is accounted for in the current analyses.

This is a great point. The denominators of the ratios in **Eqs. (4)-(5)** (originally Eqs. (6)-(7)) give the Fisher information under perfect observation of infections and hence describe the intrinsic uncertainty induced by stochastic infection counts. These perfect Fisher information terms are proportional to the total infectiousness and fall with infections, reflecting how we intrinsically lose the ability to estimate growth rates and reproduction numbers in these settings. The numerators of these ratios include the intrinsic noise as well as the additional impact of the uncertainties from surveillance imperfections. Consequently, the Fisher ratios have maximum values of 1 when there is no surveillance noise, but how well we can estimate transmissibility even at a ratio of 1 depends on the intrinsic noise. We have added explanatory text around these equations to better explain how they account for both intrinsic and surveillance noise.

4. The legend for Figure 2 could be more detailed to improve clarity. For example, it would be helpful to define the terms "supercritical R" and "subcritical R" and clarify the corresponding y-axis labels. Additionally, I found it difficult to interpret the various ribbon colors overlaying newly infected cases based solely on the current legend. A more detailed explanation would help clarify their meaning. The same comment applies to Figures 3 and 4 to ensure that their messages are conveyed.

For **Fig 2** we have removed the terms supercritical and subcritical from the legend (as they were only used there), explained labels and explicitly referenced all key components of the figure panels, including explaining colours and ribbons. Note that the colours of the cases do not have any meaning other than to indicate different realisations. By ribbons we mean the credible intervals around the R estimates. We have similarly improved legends for **Figs 3-4** and enhanced figure resolution. These amendments also reflect our responses to Reviewer 2 (points 2 and 4) and hopefully make the figures clearer and more informative.

5. I believe the exponential growth rate (which was used in the method with case counts) and reproduction numbers could be considered somewhat interchangeable (especially in the early phase when we can assume there is negligible population-level immunity). Given this, I am curious about the rationale behind treating these two concepts differently in the analysis. Could the authors clarify the distinction they made and why they chose to approach them separately?

While we agree that there is overlap between growth rates and reproduction numbers, they offer differing perspectives on (and assumptions about) transmissibility e.g., see debates in [3–5], can be used in distinct ways to assess interventions [4] and can both be computed from diverse data streams in a multitude of ways. More importantly, they form the two main metrics of transmissibility that are widely used to inform decisions and raise disease awareness (e.g. the UK government reported both R and r during COVID-19). Consequently, we included both measures to ensure we covered all key decision-making indicators and also to demonstrate (which as far as we can tell is a novel result) that the impact of surveillance noise on the Fisher information ratio is exactly the same, even though the Fisher information values themselves are different (even when there is no surveillance noise). Importantly, across [3–5] it has been shown that growth rates can sometimes be more robust (if less informative) than reproduction numbers (e.g., if generation times are misspecified). We show that this robustness does not help with surveillance noise. We have worked some of this rationale into text around **Eq. (5)**.

6. It would be beneficial if the authors discussed potential strategies or directions for further studies that could address the asymmetry issues highlighted in the study. This would provide readers with a broader perspective, not only on how outbreak responses may fail but also on ways to mitigate such failures.

Thanks, we have expanded paragraphs in the Discussion to outline some potential ways that we might circumvent these asymmetry issues, which make it harder to respond to emerging or resurging outbreaks in a timely and effective manner. We envision two possible solutions. First, where possible we should supplement existing disease surveillance frameworks with auxiliary data sources such as sentinel species monitoring and web-based digital streams. These can provide early warning signals that anticipate epidemic growth and overcome the asymmetric bottleneck that would occur if responses were only based on case data. However, this requires coordinated and potentially expensive sampling and web scraping that may not always be possible or may be out of the budgets of some locations. Second, we recommend, particularly when budget issues or surveillance capacities are limiting, making precautionary or proactive decisions during growing outbreaks. This increases sensitivity (e.g., perhaps the threshold number of cases for initiating action is lowered) and so overcomes the expected bottleneck from surveillance imperfections (though false alarms may increase).

7. [Minor] I suggest that the authors explicitly define the reproduction number in the Results section. In this context, the term “effective reproduction number” may be the right choice?

Thanks, in that Results section we now explicitly define that we refer to effective reproduction numbers. We also highlight that these are equivalent to the basic reproduction number during the early growth stages of the epidemic.

Reviewer #2:

The manuscript is well-written and, to the best of my knowledge, the asymmetric contribution of noise sources for the timely intervention on epidemics represents a novel finding for the field of mathematical epidemiology. In addition, I find the study very timely and the results reported here are relevant, as addressing the inherent constraints limiting the design and implementation of control strategies improves our preparedness and response to epidemic

scenarios. The manuscript is sound and combines results from simulations, using different frameworks, with some analytical insights obtained from a simplistic mean-field model. Because of all these factors, in my opinion, the study has potential to be published in Communications Physics. Nevertheless, I have some major comments on the manuscript that I would like the authors to address:

Thanks for the constructive critique, which has improved and helped clarify the study.

1. The manuscript should be restructured to be focused on the main findings derived from the study. Namely, the manuscript starts with an exhaustive description of Bayesian decision theory. This section is informative and useful to become familiar with a probabilistic grounded decision theory. However, most of the contents explained there are not essential for the reader to understand the results reported in the manuscript. For this reason, I would encourage the authors to move this section as a subsection of the Methods section.

Thanks for this helpful suggestion. We have now shortened this section, porting the technical and less relevant components to a new Methods subsection as suggested. We did not fully remove this Results subsection since it is important for the proceeding results to establish that the reason we focus on these decision-thresholds is because they can be directly and broadly mapped to various cost trade-offs, even when costs are not known precisely. This is not only a key point about the relationship between thresholds and uncertain costs but also provides mathematical evidence that our subsequent investigations are generalisable. We have further amended text around the Results (see response to point 1 of Reviewer 1) to better explain our findings and have expanded the Discussion on the actions that could be taken to ameliorate the implications of our findings. Hopefully, these changes in combination make the paper more accessible, clear and better focussed.

2. The authors correctly identify the origin of the asymmetry but the explanation of the compensation of both noise sources is not strictly correct. I agree that Eq. (4) holds for the exponential growth of epidemics, being r_0 the growth rate. In contrast, the declining phase of epidemics does not always have to be strictly exponential with a fixed exponent, especially when $R \leq 1$. If exponential, the decay rate is shaped by R rather than by r_0 and should incorporate the fraction of susceptible population. Therefore, it is not reflected by the initial growth rate r_0 , as suggested in Eq (5). To provide a more rigorous explanation, I would recommend the authors to frame the explanation with general rates r for the exponential growth and decay of epidemics.

Thanks for highlighting this. We agree and have now reframed the explanation and derivation in terms of general growth rates r and the accompanying R and emphasise that these assume constant parameters that are effective rates and reproduction numbers. We also note that if susceptible depletion or other time-varying transmissibility changes are important, then these quantities are interpreted as the average value over the period of interest. Given a general growth rate $r(t)$ and initial condition I_{ic} , we can substitute for an average \bar{r} as follows.

$$I_t = I_{ic} e^{\int_{t_{ic}}^t r(s) ds} \approx I_{ic} e^{\bar{r} \Delta t} \text{ with } \bar{r} = \frac{1}{\Delta t} \int_{t_{ic}}^t r(s) ds \text{ and } \Delta t = t - t_{ic}.$$

We can relate an averaged effective reproduction number, approximately, via the SIR/SEIR expressions in that equation or more broadly by using the Euler-Lotka relationships in [6]. We amend the text to include some of these points (also around the new **Eq. (5)**).

3. The section on use of transmissibility as an indicator to enforce/lift interventions interrupts the natural flow of the manuscript. While Figures 1, 3, 4 are generated using simple/complex criteria accounting for case numbers, Figure 2 reflects what happens with the Fisher information for an epidemic in the growth/declining phase. To introduce this figure, the authors motivate the fact that sometimes the indicator used for policymaking is the effective reproductive number. While using this criterion is useful for intermittent policies (producing oscillatory dynamics), it is not useful for the first epidemic wave as the effective reproduction number can only decrease over time (policies should always enforced from the beginning). For this reason, I only find this section useful once the first epidemic curves with epidemic dynamics have been shown.

Moreover, current Figure 2 does not illustrate the practical implications of the asymmetry found in the previous section beyond stressing the fact the Fisher information values from the exact and the noisy models are closer during the epidemic declining phase. To solve both issues, I will move the section using the reproduction number to the end of the manuscript and complement the results shown in Figure 2 showing another panel with a histogram of the delay in enforcing/lifting the control policies using the effective reproduction number R as the information for decision-making. In that sense, it would also be interesting to check whether underreporting also counteracts lagged case reports when using R .

Thanks for highlighting this. Our goal here was just to generically explain that even under other popular indicators for decisions i.e., time-varying growth rates r and reproduction numbers R , there is still an important asymmetry. At the beginning of the first wave, the basic reproduction number, which is important for initial interventions, equals R and so we believe our analysis does provide some value for both the first and subsequent waves. Further, other factors such as population behaviours and spatial heterogeneities can mean that R varies even in the first wave as in [7] or when interventions induce generation time changes as in [8]. Even if these issues do not exist, it is not obvious that the optimal action is to enforce policies from the start as policymakers may wait for data (the evidence for H_t in our framework) to accumulate to get more precise estimates of R for justifying strong or costly actions (η in our framework) [9].

However, we take the reviewer's concerns about making the flow clearer and linking better to practical implications. Towards that end, we have included a panel within **Fig 2** with the suggested histogram of delays in decisions in growing and waning phases from infections versus cases. Here actions are triggered when the cumulative probability of the estimated R being above or below 1 exceeds a cost threshold. This panel serves two purposes. First, it confirms that the asymmetry holds but the effect in this example seems smaller (we are ~ 3.5 days slower to respond during growth phases) than in **Fig 1**. Second, this panel underscores why we chose to specify the asymmetry in R (and r) via the Fisher information. Making decisions using case thresholds is clear because cases are recorded data. But R is an estimated statistic and so its practical uncertainty and hence the asymmetrical effect size also depend on methodological choices (e.g., Fig S1 in [10] plots different R estimates from the

same case data). The Fisher information is more objective and bounds performance for all unbiased, asymptotic estimators.

We also now include more linking text between sections (see response to point 1 of Reviewer 1) to improve flow and provide better rationale for the manuscript organisation. This structure is i) we define optimal decision problems in terms of evidence for H_1 and a cost threshold η , ii) we explore fundamental decisions with cases as evidence, iii) we investigate fundamental decisions with R estimate precision as evidence and iv) we bring all this together using MPC algorithms that are predictive, utilise both case data and R estimates in decision-making and explicitly consider cost thresholds. Previously we had not made clear enough that R estimation was a part of our predictive algorithms. We have now included text on this in the Methods and in the Results and hopefully our manuscript organisation is now more logical.

4. Resolution of Figures 3 and 4 should be improved.

We have now included full resolution versions of these figures as separate files and embedded them in the text as well, though MS Word may result in reduced resolutions. Hopefully, the new figures are clearer and more interpretable.

5. Minor comments:

Use another color different from the tone chosen for the 'green' histograms in Figures 3 and 4 as it is difficult to identify the color. Also use thicker lines for the histogram to be more visible.

We have implemented these changes in the revised manuscript.

λ should be defined in the main text.

Thanks, this has been included now.

I think that colors of the histograms corresponding to underreporting and delayed reports are inter-exchanged in Figure 1.

Thanks for this incisive observation. The colours are correct but on inspection we realised we miswrote the Δ_{wane} expression (we gave the formula for $-\Delta_{\text{wane}}$). This has been rectified in the new **Eq. (3)** and now corresponds correctly to the histograms.

Bibliography

1. Parag KV, Donnelly CA, Zarebski AE. Quantifying the information in noisy epidemic curves. *Nat Comput Sci.* 2022;2: 584–594. doi:10.1038/s43588-022-00313-1
2. Karlinsky A, Kobak D. Tracking excess mortality across countries during the COVID-19 pandemic with the World Mortality Dataset. *Elife.* 2021;10. doi:10.7554/eLife.69336
3. Parag KV, Thompson RN, Donnelly CA. Are epidemic growth rates more informative than reproduction numbers? *J Royal Statistical Soc A.* 2022; doi:10.1111/rssa.12867
4. Dushoff J, Park SW. Speed and strength of an epidemic intervention. *Proc Biol Sci.* 2021;288: 20201556. doi:10.1098/rspb.2020.1556
5. Pellis L, Scarabel F, Stage HB, Overton CE, Chappell LHK, Fearon E, et al. Challenges in control of COVID-19: short doubling time and long delay to effect of

- interventions. *Philos Trans R Soc Lond B, Biol Sci.* 2021;376: 20200264.
doi:10.1098/rstb.2020.0264
6. Wallinga J, Lipsitch M. How generation intervals shape the relationship between growth rates and reproductive numbers. *Proc R Soc B.* 2007;274: 599–604.
 7. Parag KV, Cowling BJ, Donnelly CA. Deciphering early-warning signals of SARS-CoV-2 elimination and resurgence from limited data at multiple scales. *J R Soc Interface.* 2021;18: 20210569. doi:10.1098/rsif.2021.0569
 8. Ali ST, Wang L, Lau EHY, Xu X-K, Du Z, Wu Y, et al. Serial interval of SARS-CoV-2 was shortened over time by nonpharmaceutical interventions. *Science.* 2020;369: 1106–1109. doi:10.1126/science.abc9004
 9. Thompson RN, Gilligan CA, Cunniffe NJ. Control fast or control smart: When should invading pathogens be controlled? *PLoS Comput Biol.* 2018;14: e1006014. doi:10.1371/journal.pcbi.1006014
 10. Steyn N, Parag KV. Robust uncertainty quantification in popular estimators of the instantaneous reproduction number. *medRxiv.* 2024; doi:10.1101/2024.10.22.24315918

Responses to Reviewers: COMMSPHYS-25-0151

We thank both reviewers for their constructive and insightful comments. We list our responses to the last comments from Reviewer 2.

Reviewer #2:

The authors have addressed most of my concerns and the revised version of the manuscript is much clearer and present a more coherent flow that improves its readability. While I am overall satisfied with the changes introduced, I think that the section "Responding to epidemic growth and decline using reported incidence" still needs to be rewritten to ease the understanding of Equations 2 and 3. Before including any mathematical expression, I would recommend the authors to explicitly mention that cases suffer from underreporting and delay in the reports and define the two mathematical parameters used to account for both factors.

Thanks for this suggestion, we have made exactly these changes in the main text.

Likewise, the authors should explicitly state that the rates r included in Eq.(3) are different, as in the growth phase corresponds to r_0 while in the declining phase corresponds to the time average of the growth rate over the window of interest (as now noted in the manuscript). Moreover, the comment on τ_p before Eq.(3) hinders understanding the fact that this delay is not an input parameter but arises from the equations as a result of underreporting. To avoid confusion, I would omit that mention and define τ_p explicitly in Equation 3 rather than ρ , as the latter is an input parameter to model underreporting.

Thanks, we have also made exactly these suggested changes now.